# Characterization of Potential Melanoma Predisposition Genes in High-Risk Brazilian Patients

**DOI:** 10.3390/ijms242115830

**Published:** 2023-10-31

**Authors:** Bianca Costa Soares de Sá, Luciana Facure Moredo, Giovana Tardin Torrezan, Felipe Fidalgo, Érica Sara Souza de Araújo, Maria Nirvana Formiga, João Pereira Duprat, Dirce Maria Carraro

**Affiliations:** 1Skin Cancer Department, A.C. Camargo Cancer Center, São Paulo 01529-001, Brazil; bianca.sa@accamargo.org.br (B.C.S.d.S.); luciana.moredo@accamargo.org.br (L.F.M.); joao.duprat@accamargo.org.br (J.P.D.); 2Clinical and Functional Genomics Group, International Research Center/CIPE, A.C. Camargo Cancer Center, 440 Taguá St., São Paulo 01508-010, Brazil; giovana.torrezan@accamargo.org.br (G.T.T.); felipe.fidalgo9@gmail.com (F.F.); erica.sara.araujo@gmail.com (É.S.S.d.A.); 3National Institute of Science and Technology in Oncogenomics and Therapeutic Innovation, 440 Taguá St., São Paulo 01508-010, Brazil; 4Oncogenetics Department, A.C. Camargo Cancer Center, São Paulo 01509-900, Brazil; nirvana.formiga@accamargo.org.br

**Keywords:** familial melanoma, melanoma predisposition, melanoma susceptibility, germline pathogenic variants, multiple primary neoplasms

## Abstract

Increased genetic risk for melanoma can occur in the context of germline pathogenic variants in high-penetrance genes, such as *CDKN2A* and *CDK4*, risk variants in low- to moderate-penetrance genes (*MC1R* and *MITF*), and possibly due to variants in emerging genes, such as *ACD*, *TERF2IP,* and *TERT*. We aimed to identify germline variants in high- and low- to moderate-penetrance melanoma risk genes in Brazilian patients with clinical criteria for familial melanoma syndrome. We selected patients with three or more melanomas or melanoma patients from families with three tumors (melanoma and pancreatic cancer) in first- or second-degree relatives. Genetic testing was performed with a nine-gene panel (*ACD*, *BAP1*, *CDK4*, *CDKN2A*, *POT1*, *TERT*, *TERF2IP*, *MC1R,* and *MITF*). In 36 patients, we identified 2 (5.6%) with germline pathogenic variants in *CDKN2A* and *BAP1* and 4 (11.1%) with variants of uncertain significance in the high-penetrance genes. *MC1R* variants were found in 86.5%, and both red hair color variants and unknown risk variants were enriched in patients compared to a control group. The low frequency of germline pathogenic variants in the high-penetrance genes and the high prevalence of *MC1R* variants found in our cohort show the importance of the *MC1R* genotype in determining the risk of melanoma in the Brazilian melanoma-prone families.

## 1. Introduction

Melanoma is a multi-factorial skin cancer and emerges from an interaction between environmental exposure and genetic susceptibility. Ultraviolet radiation (UVR) is the major known etiologic agent associated with melanoma [1]. The risk determined by UVR depends on geographic latitude, cumulative and exposure pattern, and history of sunburn, which is related to specific phenotypic features such as fair skin, red/blond hair, light-colored eyes, propensity to burn, multiple nevi, and presence of atypical nevi [1,2]. The genetic basis of melanoma development stems from random acquired mutations and the accumulation of genomic changes within melanocytes. At the same time, melanoma susceptibility is also determined by the presence of heritable germline variants [3]. Patients with a personal and or a family history of melanoma are the ones at highest risk for developing the disease [2]. Approximately 5–10% of melanoma cases occur in a familial context, which implies that an inherited germline pathogenic variant (GPV) in cancer-predisposing genes, a shared environmental exposure, or both concur with melanoma pathogenesis [4,5]. Outside the familial context, multiple primary melanomas (MPM) might also be related to genetic factors such as germline de novo pathogenic variants in high-penetrance genes and multiple variants in moderate-penetrance genes [6].

*Cyclin-dependent kinase inhibitor 2A* (*CDKN2A*) was the first high-penetrance melanoma gene identified and encodes two distinct proteins (p16 and p14ARF), both acting in cell cycle regulation [7]. In general, *CDKN2A* GPV is found in 10 to 40% of melanoma-prone families [8,9] and is associated with an increased number of family members affected early age at melanoma diagnosis, MPM, and pancreatic cancer [10].

The second melanoma predisposition gene described was *CDK4*, an oncogene that also plays an important role in cell cycle regulation. The phenotype of *CDK4*-mutated families is indistinguishable from *CDKN2A*-mutated ones [11]. Only 20 families have been identified with GPVs in the *CDK4* gene to date [12,13].

Melanocortin 1 receptor (*MC1R*) and microphthalmia-associated transcription factor (*MITF*) are considered moderate-penetrance melanoma genes. *MC1R* is crucial in the control of human pigmentation and is a highly polymorphic gene. *MC1R* variants are associated with an increased risk of developing melanoma, especially the R variants, which are highly associated with red hair color phenotype [14]. Melanoma risk determined by *MC1R* R variants varies from two to three times per R allele and is higher in the familial context when compared with the general population [14]. *MITF* regulates melanocyte development and differentiation. The p.E318K variant has been associated with high nevi count, fair skin, and non-blue eye color and predisposes to both melanoma and renal cell carcinoma [14,15].

More recently, other high-penetrance melanoma genes have been identified, such as breast cancer 1 (BRCA1)-associated protein 1 (*BAP1*), with definitive evidence for association with melanoma and other tumors and other potential melanoma-predisposing genes related to telomere maintenance–protection of telomeres 1 (*POT1*), *ACD*, *TERF2IP,* and telomere transcriptase reverse (*TERT*) [16,17,18], which still lack validation from other studies. To date, these aforementioned genes have not been studied in Brazilian patients. Along with *CDK4*, they correspond to less than 3% of melanoma-prone families in the studied populations around the world; nevertheless, the majority of cases remain without known genetic etiology [4,14,18].

Familial clustering of melanoma and MPM cases are referred to as familial or hereditary melanoma. In countries with low to medium incidence of melanoma, clinical criteria for familial or hereditary melanoma include patients with two primary melanomas and families with two melanoma cases and/or one pancreatic carcinoma and one melanoma, in first- or second-degree relatives (“rule of two”) [19]. In areas with moderate to high melanoma incidence, the recommendation is to consider the “rule of three”—patients with three primary melanomas and families with three melanoma cases [19].

In Brazil, the incidence of melanoma varies widely according to the different geographic regions, mainly due to the population’s ancestry, latitude, and UVR emission during the summer months [20,21]. The World Health Organization (WHO) shows, for 2020, an estimated melanoma incidence rate in Brazil of 2.7 (female) and 3.1 (male) per 100,000 [22]. The Brazilian National Cancer Institute estimates for 2020 a melanoma incidence rate that varies from 1.36 to 6.78 per 100,000 for the southeast region and 3.53 to 10.05 per 100,000 for the southern region, the two Brazilian regions with the highest melanoma incidence, achieving incidence rates similar to European countries such as Spain [22,23]. 

Data regarding the frequency of germline mutations in the *CDKN2A* gene for Brazilian individuals fulfilling clinical criteria for familial melanoma are scarce and differ according to the region studied and the adopted criteria, varying from 0% to 14% [21,24,25,26]. In a study from southeast Brazil, a mutational rate of 4.5% in *CDKN2A* was observed in a cohort of 22 probands [24]. Another study from south Brazil found 0% (0/33) of *CDKN2A* GPV in unrelated patients selected according to the “rule of two” [21]. In a previous study of our institution, 14% of GPV carriers in *CDKN2A* were detected in a cohort of 59 unrelated patients from the southeast region of Brazil [25]. The clinical criteria used were also the “rule of two”. However, when it was considered that families presenting both MPM and multiple relatives were affected, the mutation rate of *CDKN2A* rose to 36% [25].

Besides *CDKN2A*, other genes potentially associated with familial melanoma have also been investigated in Brazilian melanoma-prone families, such as *CDK4*, *MITF,* and *TERT* promoter variants [24,25,27]. No pathogenic variant in *CDK4* was identified so far [21,24,25]. Both *TERT* promoter mutation at −57 bp from the start site and the *MITF* p.E318K variant, which are, respectively, high- and moderate-penetrance variants, were also assessed in 48 unrelated probands negative for *CDKN2A* GPV, detecting only one patient with the *MITF* GPV p.E318K [27]. 

The purpose of this study was to perform a mutational analysis of the seven candidate melanoma susceptibility genes (*ACD*, *BAP1*, *MC1R*, *MITF*, *POT1*, *TERF2IP*, and *TERT*) in high-risk melanoma patients (familial melanoma and MPM) from southeastern Brazil, besides the *CDKN2A* and *CDK4* genes. Additionally, we aimed to verify the association among patient’s GPV status, their clinical features, pathological characteristics of their tumors, and the presence of other malignancies, both in patients and their relatives.

## 2. Results

A total of 37 patients (MH1-MH12 and MH14-MH38), 19 women and 18 men were enrolled in this study, all, except MH31, with three or more melanomas or two or more first- or second-degree relatives with melanoma or pancreatic cancer (“rule of three”). MH31 had only one first-degree relative with melanoma and a personal history of cutaneous melanoma, kidney tumor, GIST (gastrointestinal stromal tumor), and multiple skin carcinomas. Additionally, his twin brothers had both melanoma and kidney tumors.

Regarding the series, the mean age at diagnosis was 40.1 years (15–69 years), with a significant prevalence of phototypes I and II (89.2%), blue and green eyes (54%), and red and blond hair (59.4%). Sunburn in childhood was reported in 89.2% of the cases, and a high density of freckles on the back was found in 73% of the probands. Approximately 50% of the cases had a total nevus count greater than 100, and 40% fulfilled the AMS criteria. The 37 probands had a total of 148 primary melanomas, with a predominance of the superficial spreading histopathological subtype (85.8%), a higher incidence on the trunk (52.7%), and a higher prevalence of in situ and thin melanomas (96.7%). Nevus-associated melanoma was found in 50.7% of the cases, but in 16.9% of the tumors, this information was not available.

A total of 26 individuals (70.3%) were classified as FM and 11 (29.7%) as MPM. In FM patients, 3 (11.5%) had a family history of pancreatic cancer, 12 (46.2%) had more than one relative affected by melanoma, 6 (23.15%) had relatives with multiple primary tumors (≥2 melanomas), and 21/26 (80.8%) were also diagnosed with multiple primary tumors (11 patients with 2 melanomas and 10 patients with 3 or more melanomas). Other tumors found in FM probands were non-melanoma skin cancer (7/26), thyroid (3/26), breast (2/26), kidney (1/26), and prostate (1/26). The tumors found in FM relatives were breast (6), non-melanoma skin cancer (5), lung (4), myeloma (3), kidney (3), colon (3), lymphoma (2), and thyroid (2).

In MPM patients, 5/11 (45.5%) had three melanomas, and 6/11 (54.5%) had four or more melanomas. Another tumor found in MPM probands was non-melanoma skin cancer (4/11), and in their relatives were breast (4), non-melanoma skin cancer (2), thyroid (1), lung (1), lymphoma (1), mesothelioma (1), gastric cancer (1), colon (1), and prostate (1).

Compared to FM, MPM patients were younger (35.6 vs. 43.5, *p* = 0.047) and showed a higher median number of total nevi (145 vs. 74, *p* = 0.04) (Table 1). The anatomical site distribution of tumors was different between the two groups: the incidence of melanomas located in the head/neck and trunk was higher in the MPM group, and the incidence of melanomas in limbs (both upper and lower) was higher in FM patients (*p* = 0.04). No differences were found concerning phenotype (eye and hair color, phototype, and AMS), freckles density, history of sunburn, and other tumor pathological features.

Regarding genetic analysis, we included the 36 patients who had three or more primary melanomas, or at least three melanomas diagnosed in the family, including melanoma and pancreatic cancer in relatives (the proband MH31 was not considered only for these analyses). Overall, we detected five patients with potentially relevant genetic variants in the high-penetrance genes studied. Their pedigrees are shown in Figure 1. Two patients (2/36—5.6%) showed GPV in *CDKN2A* and *BAP1* genes, and four patients (4/36–11.1%) showed prioritized variants of uncertain significance (VUS) in *ACD*, *POT1,* and *TERT* genes. Detailed information on all variants identified is listed in Table 2.

The patient harboring the GPV in the *CDKN2A* gene (MH11) was also detected with one of the *ACD* VUS and a non-RHC variant of *MC1R* (r/0). The patient has fair skin, a history of sunburn both in childhood and adulthood, total nevi count < 100, no AMS phenotype, a personal history of multiple melanomas early diagnosed (30 years old), a mother also with multiple melanomas, and a family history of pancreatic, breast and colon cancer (Figure 1A). The p.Glu419Lys variant in the *ACD* gene is not described in ClinVar and is classified as LB by Varsome and as VUS by Franklin. In silico analysis showed conflicting data, with a predominance of benign predictions.

The proband carrying the GPV in the *BAP1* gene (MH20) also showed an RHC variant of *MC1R* (R/0) and had multiple nevi, fair skin, and a history of sunburn in childhood, as well as multiple melanomas diagnosed at an early age. The patient does not have a family history of melanoma but has two family members diagnosed with mesothelioma. The patient’s sister, still asymptomatic, also carries the same *BAP1* variant (Figure 1B). 

The other case where *ACD* VUS was found (MH32) has a personal history of multiple melanomas and a family history of melanoma and breast cancer (Figure 1C) and showed a non-RHC variant of *MC1R* (r/0), in addition to fair skin, light hair, and a history of childhood sunburn. The p.Ser321Leu *ACD* variant is classified as VUS by ClinVar and Franklin and as LB by Varsome, presenting conflicting data from in silico analysis with the majority of pathogenicity predictors indicating benignity.

The patient that carries the *TERT* VUS (MH28) shows a high density of freckles, a personal history of multiple melanomas and colloid thyroid goiter, and a family history of melanoma with multiple family members affected (Figure 1D). A variant of unknown risk of *MC1R* (u/0) was identified. The p.Glu441del *TERT* variant has conflicting data both in ClinVar and pathogenicity functional studies. Twelve laboratories classify it as LB/B, three as VUS, and one as P. The Varsome tool classifies this variant as VUS and Franklin as B.

The proband where the *POT1* VUS was detected (MH16) also showed two *MC1R* variants (R/u), fair skin, light hair, a history of sunburn in childhood, a personal history of six primary melanomas and thyroid cancer, and a family history of melanoma (Figure 1E). The p.Phe106Leu *POT1* variant has not been described in ClinVar and is classified as VUS by Varsome and Franklin, showing conflicting data from in silico analysis with the majority of pathogenicity predictors indicating pathogenicity.

Of all 37 probands, 86.5% (32/37) showed at least one *MC1R* variant. The R alleles (R151C, R160W, D84E, D294H, and R142H) were detected in 54.1% of the patients (20/37), with the R151C and R160W being the most prevalent variants (27% and 18.9%, respectively). Fifteen patients (40.5%) showed r alleles (V60L, V92M, I155T, and R163Q). V60L was the most prevalent r variant, found in 24.3% of the probands. Only five patients were detected with u variants (13.5%). Nine patients carried two allelic variants. Probands were classified as R/R, R/r, R/u, R/0, r/r, r/u, r/0 or u/0 according to the allelic combination. Two patients were detected with one homozygote R and r variant and were also classified as R/R and r/r, respectively. Five probands were *MC1R* wild type and were classified as 0/0. *MC1R* allelic data are shown in Table 3. No differences were found between FM and MPM patients concerning the *MC1R* variants detected (*p* = 0.748 R variants, *p* = 0.080 r variants, and *p* = 1.00 u variants).

The R151C variant was associated with blue/gray eyes (*p* = 0.042) and red/blond hair (*p* = 0.006), where 81.8% of carriers had blue or gray eyes, and all carriers were blond or redhead. No associations were found between the *MC1R* variants and sex, age at diagnosis, other phenotypic characteristics, number of primary melanomas, or tumors histopathological characteristics.

We observed an important increase in the R variants in our cohort compared to the ABraOM control group (OR 5.94, 95% CI 3.44–10.26, *p* < 0.001) (Table 4). The R151C (O.R. 9.25, 95% C.I. 4.34–19.74, *p* < 0.001) and R160W (O.R. 4.94, 95% C.I. 2.08–11.74, *p* < 0.001) variants were the most important. There was no difference when comparing the r variants, but the u alleles were significantly more frequent in our cohort (O.R. 10.25, 95% C.I. 2.40–43.75, *p* = 0.002). 

## 3. Discussion

This study characterizes a cohort of 37 high-risk melanoma patients (probands with strong family history and multiple primary melanoma individuals) according to phenotype, histopathological tumor features, pedigree, and mutational status of the main melanoma susceptibility genes. 

Our patients had an earlier age at their first melanoma diagnosis (mean of 40.1 years) than the general population, with a higher prevalence of thin melanomas (96.7%), predominance of the superficial spreading histological subtype (85.8%), and a large number of patients with a history of childhood sunburn (89.2%), as described in other studies that characterized FM patients [28,29]. Patients with multiple melanomas alone without a family history of melanoma (MPM) had a significantly lower average age at diagnosis of the first melanoma than those with a family history of melanoma, but no data were found in the literature to corroborate or justify this finding. The frequency of melanomas located in limbs was higher in FM patients. Although some studies showed the trunk as the most frequent location of cutaneous melanoma in both sexes, a recent study has presented a higher frequency of melanomas in females in the limbs [30], which could justify our findings since the FM group showed a higher frequency of females (Table 1).

The frequency of GPVs found in the high-penetrance genes studied was low (5.6%—2/36; 4.0% for FM group—1/25 and 9.1% for MPM group—1/11), despite the selection criteria adopted (“rule of three”) and the number of melanomas found in the probands and their relatives. These results differ from the literature and previous Brazilian studies (0–14%), including one from our cancer center (36%—4/11, when considering FM + MPM) [9,31,32,33]. This difference is likely to be due to the overall small sizes of ours and previously described cohorts.

Regarding the clinical phenotypes of GPV carriers, the case that shows the GPV in *CDKN2A* (MH11) exhibits the typical phenotype described for *CDKN2A* germline mutations—early age at diagnosis, multiple primary melanomas, multiple family members affected, and association with pancreatic cancer [9,10]. The proband carrying the GPV in the *BAP1* gene (MH20) was also diagnosed with cutaneous atypical spitzoid tumors, BIMTs (*BAP1*-inactivated melanocytic tumors), which are considered clinical markers of *BAP1* germline mutation. The detailed description of the clinical, dermoscopic, confocal microscopy, and histopathological aspects of the skin tumors of this patient (melanomas and BIMTs), in addition to the genetic aspects, have been published recently [34]. 

The *POT1* VUS found in proband MH16 is located in the DNA-binding domain 1 (OB1), in which other pathogenic missense variants have been described [35]. Recent studies identified *POT1* germline variants in melanoma-prone families with cases of thyroid cancer, as seen in this case [36,37]. This proband showed two *MC1R* variants (R/u). It is described that patients with two *MC1R* variants have a higher melanoma risk compared to those with single variants [38]. 

Many negative cases for the high-penetrance genes studied have multiple primary melanomas, and sometimes also multiple family members affected or relatives with multiple melanomas, featuring high-risk melanoma-prone families of great interest in deepening the genetic study by extension of the genetic panel according to the other tumors found in the family and whole-exome sequencing. In this sense, recent studies have attempted to identify new melanoma-predisposing genes through comprehensive genomic studies, but although interesting candidates have been recognized, no strong validated new genes have been described so far [39]. It is also possible that in some of these families, the increased risk for melanoma may be related to the *MC1R* genotype combined with environmental exposure since melanin synthesis is related to melanoma progression. *MC1R* risk variants, especially R variants, determine an imbalance between eumelanin and pheomelanin, with an increase in the latter. Pheomelanin provides a pro-oxidative cellular environment and induces DNA damage, contributing to melanomagenesis [40].

A recent study carried out in Turkey [41] corroborated the importance of the *MC1R* genotype in countries with a low incidence of melanoma and mixed population in determining the melanoma risk.

The main limitation of our study was the small size of our cohort, determined by funding constraints, supplying us with no sufficient statistical power in some analysis, especially when comparing the two subgroups FM × MPM and *MC1R* variants assessment. This limited size is an important factor that impacts the generalizability of our findings. Moreover, the absence of a comparison with a control group for the other genes beyond *MC1R*, due to the limited number of positive patients, hindered our ability to definitively establish the overall role of these genes in melanoma risk in the Brazilian population. Still, for *MC1R*, our comparisons from our cohort to a population-matched control group showed an increased frequency of R and u variants in the patients’ group. 

Our patients showed a high prevalence of *MC1R* variants (86.5% total; 88.5% in FM and 81.8% in MPM), with the R151C (R) and V60L (r) being the most frequent variants detected (27.3% and 24.3%, respectively). A recent polled analysis showed that the presence of any *MC1R* variant confers a 60% higher risk for cutaneous melanoma to carriers when compared to noncarriers [42]. RHC variants were identified in over 50% of all probands, with a significant increase when compared to the control group, corroborating their high prevalence in melanoma patients [43]. In our study, the R151C variant was associated with light hair (red and blond) and blue/gray eyes, supporting the role of RHC variants in the determination of light phenotypic complexion [44]. 

An interesting case was the MH29 proband (Table 2), who developed multiple primary melanomas diagnosed at an early age and had a first-degree relative also with multiple melanomas. No variant in the high-penetrance genes studied was found, but two RHC variants of the *MC1R* gene (R/R) could explain, partly, the high-risk profile for melanoma seen in this proband. A recent study reported that *MC1R* genotype and nevi number concur synergistically to melanoma risk, and the R/R genotype combined with high nevi count results in a deeply high-risk outline for melanoma [45].

The MH01 proband (Table 2) also had multiple primary melanomas, and the first tumor was diagnosed before the age of 40. His brother also has multiple primary melanomas in addition to non-melanoma skin cancer and thyroid carcinoma. The proband showed no variant in the high-penetrance genes studied. Unlike the proband MH29, this patient presented only one u variant (p.Gln23Ter—unknown risk) of *MC1R* (u/0). Despite presenting an unknown risk, the p.(Gln23Ter) variant is a loss-of-function (LoF) variant, and *MC1R* LoF variants have already been associated with the risk of MPM [46]. 

Moreover, this group of *MC1R* variants (u alleles) showed a higher prevalence in our cohort when compared with the control group and may characterize high-risk melanoma patients in the Brazilian population. 

## 4. Materials and Methods

### 4.1. Patients’ Selection

Melanoma patients recruited between 2016 and 2019 from the Familial Melanoma Clinic at the Skin Cancer Department of A.C. Camargo Cancer Center, São Paulo, Brazil, with 3 or more melanomas or 2 or more first or second-degree relatives with melanoma or pancreatic cancer (“rule of three”) were included in the study. All participants provided written and informed consent, and this study was approved by the ethics committee of the A.C. Camargo Cancer Center (2076/15).

A standardized questionnaire containing phenotypic characteristics, patients’ demographic data, and tumor features was used, including sex, age at diagnosis, eye and hair color, skin phototype, total nevi count, presence of atypical mole syndrome (AMS) phenotype, freckles density, history of sunburn, Breslow thickness, tumor location, histological subtype, tumor-associated nevus, staging, proband and relatives’ number of melanomas, and proband and relatives’ presence of other tumors.

Patients were classified as FM (familial melanoma = family history of melanoma or pancreatic cancer) or MPM (multiple primary melanomas = without family history of melanoma or pancreatic cancer).

### 4.2. Germline Genetic Analysis

Genomic DNA was obtained from blood leucocytes or saliva and then was subjected to a library preparation protocol according to an Ion AmpliSeq™ Library Kit 2.0 and sequenced with the Ion Proton Platform (Thermo Fisher Scientific, Waltham, MA, USA). The entire coding region of nine melanoma predisposition genes (*ACD*, *BAP1*, *CDKN2A*, *CDK4*, *MC1R*, *MITF*, *POT1*, *TERF2IP*, and *TERT*) were analyzed using a custom Ion Ampliseq™ Panel (Thermo Fisher Scientific, Waltham, MA, USA). 

The Torrent Variant Caller tool 5.0–13 (ThermoScientific) was used to perform the alignment and the variant calling. The identified variants were noted and filtered using the VarSeq software (Version 1.8, Golden Helix, Bozeman, MT, USA). Variants with the following quality criteria were selected: base coverage ≥50× and frequency of the variant allele ≥0.25. The detected variants were evaluated for their classification in databases of clinical classification of variants (ClinVar, LOVD) and population allele frequency (gnomAD, 1000G, ESP, and ABraOM) using the Varsome tool. The variants were classified according to the criteria suggested by the American College of Medical Genetics (ACMG) [47]. 

The *MC1R* variants were classified according to their association with the red hair phenotype and risk of developing melanoma [48]:

R alleles or RHC (red hair color) variants: p.D84E, p.R142H, p.R151C, p.R160W, and p.D294H—2 times increased risk for melanoma or more;

r alleles or non-RHC variants: p.V60L, p.V92M, p.I155T, and p.R163Q—1.5 times increased risk for melanoma;

u variants: p.Q23X, p.G89R, p.I264V, p.T272M, and p.S83L—unknown risk for melanoma.

### 4.3. Statistical Analysis

In assessing the association between categorical variables, the chi-square test or Fisher’s exact test was used when appropriate. To compare the two groups (FM and MPM) and the *MC1R* variants according to numerical variables, the Mann–Whitney non-parametric test was applied.

A comparison of the *MC1R* variant frequencies found in our cohort with a control group was performed using a logistic regression model to calculate odds ratios (OR) with confidence intervals (CIs) of 95%. The control group used was the database of the Brazilian Online Archive of Variants (AbraOM), which represents a cohort of elderly Brazilians (over 60 years of age) with extensive phenotyping based on the SABE study census (Health, Wellness and Aging) initiated in 1999, enclosing data from São Paulo city. As approximately 10% of the Brazilian population lives in the metropolitan region of São Paulo, this is considered a representative sample of the national population. It includes 609 individuals, 216 men and 393 women, 517 self-declared white, 156 black, 12 yellow, 2 indigenous, 40 self-declared as “others”, and 52 without definition [49].

The level of significance adopted was 5%, and statistical analyses were performed using IBM SPSS software version 25.

## 5. Conclusions

A low frequency of high-penetrance gene variants and a high prevalence of *MC1R* variants found in our cohort emphasize the likely importance of *MC1R* variants in determining melanoma risk in our population. The higher frequency of RHC and u *MC1R* variants in our cohort compared to the control group and the presence of two *MC1R* variants demonstrates that, to a certain extent, the increased melanoma risk of these individuals and families may be related to *MC1R* genotype combined to pigmentation phenotype, behavior of risk (UVR exposure) and the presence of multiple nevi. Future studies should seek to investigate how *MC1R* variants interact with other genetic and environmental risk factors in determining melanoma risk among Brazilian melanoma-prone families.

## Figures and Tables

**Figure 1 ijms-24-15830-f001:**
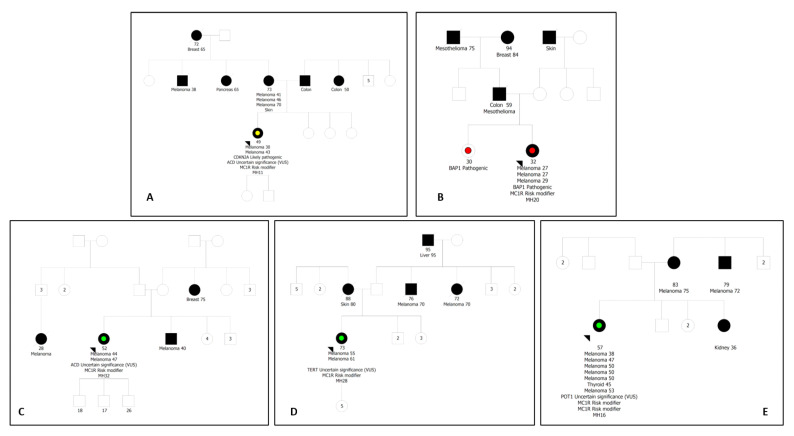
Pedigrees of the probands that showed genetic variants in the high-penetrance genes. (**A**) Pedigree of FM patient (MH11) carrying the GPV in *CDKN2A*, VUS in *ACD* gene and classified as r/0 for *MC1R*. (**B**) Pedigree of MPM patient (MH20) carrying the GPV in *BAP1* and classified as R/0 for *MC1R*. (**C**) Pedigree of FM patient (MH32) carrying the other VUS in *ACD* gene and classified as r/0 for *MC1R.* (**D**) Pedigree of FM patient (MH28) carrying the VUS in *TERT* gene and classified as u/0 for *MC1R*. (**E**) Pedigree of FM patient (MH16) carrying the VUS in *POT1* gene and classified as R/u for *MC1R*.

**Table 1 ijms-24-15830-t001:** FM × MPM—clinical characteristics and tumors histopathological features.

	FM	MPM	*p*
Age at diagnosis (years) mean	43.5	35.6	0.047
(Range)	(15–69)	(25–49)	
Total nevus count median	74	145	0.04
(Range)	(4–348)	(23–334)	
	n (%)	n (%)	
Sex	M10/F16	M8/F3	
Phototype			
I/II	22 (84.6)	11 (100)	0.296
III/IV	4 (15.4)	0	
Eye color			
Blue/green	15 (57.7)	5 (45.5)	0.748
Brown/black	11 (42.3)	6 (54.5)	
Hair color			
Red/blond	16 (61.5)	6 (54.5)	0.728
Brown/black	10 (38.5)	5 (45.5)	
Freckles density (back)			
Low	7 (26.9)	3 (27.3)	1.00
High	19 (73.1)	8 (72.7)	
Childhood sunburn			
Yes	22 (84.6)	11 (100)	0.296
No	4 (15.4)	0	
Adulthood sunburn			
Yes	13 (50.0)	9 (81.8)	0.141
No	13 (50.0)	2 (72.7)	
AMS			
Yes	8 (30.8)	7 (63.6)	0.08
No	18 (69.2)	4 (36.4)	
Nevus count			
≤100	16 (61.5)	3 (27.3)	0.122
>100	10 (38.5)	8 (72.7)	
Tumors features	n (%)	n (%)	
Breslow thickness			
In situ	57 (53.8)	19 (45.2)	
≤1 mm	46 (43.4)	21 (50.0)	0.56
>1 mm	3 (2.8)	2 (4.8)	
Histopathological subtype			
Superficial spreading	90 (94.7)	37 (97.4)	
Nodular	1 (1.1)	0 (0)	
Lentigo maligna	1 (1.1)	0 (0)	1.0
Acral lentiginous	2 (2.1)	1 (2.6)	
Others	1 (1.1)	0 (0)	
Nevus-associated melanoma			
Yes	52 (58.4)	23 (67.6)	0.46
No	37 (41.6)	11 (32.4)	
Anatomical site			
Head and neck	3 (2.8)	3 (7.1)	0.04
Trunk	50 (47.2)	28 (66.7)	
Upper limbs	18 (17.0)	5 (11.9)	
Lower limbs	35 (33.0)	6 (14.3)	

FM—familial melanoma; MPM—multiple primary melanomas without family history; M—male; F—female; AMS—atypical mole syndrome.

**Table 2 ijms-24-15830-t002:** Description of germline variants detected in known and candidate melanoma genes.

Case	Gene	Type	RefSeq (NM)	HGVS cDNA; Protein	dbSNP	MAF (gnomAD/ABraOM)	ClinVar	Varsome/Franklin	In Silico Predictors Benign (b)/Pathogenic (p)	Revel	ACMG
MH11	*CDKN2A*	missense	NM_000077.4	c.71G>C; p.(Arg24Pro)	rs104894097	0.000017/ na	P(9); PP(1)	P/LP	8 b/4 p	0.4	P
MH11	*ACD*	missense	NM_001082486.2	c.1255G>A; p.(Glu419Lys)	na	na	na	LB/VUS	11 b/1 p	0.04	VUS
MH16	*POT1*	missense	NM_015450.2	c.318T>G; p.(Phe106Leu)	na	na	na	VUS/VUS	2 b/11 p	0.71	VUS
MH20	*BAP1*	Frameshift deletion	NM_004656.3	c.1265delG; p.(Gly422Glufs*8)	na	na	na	P/LP	0 b/1 p	na	P
MH28	*TERT*	Inframe deletion	NM_198253.2	c.1323_1325delGGA; p.(Glu441del)	rs377639087	0.00164/0.00329	B/LB(12); VUS(3); P(1)	VUS/B	1 b/0 p	na	VUS
MH32	*ACD*	missense	NM_001082486.2	c.962C>T; p.(Ser321Leu)	rs374925782	0.000072/0.00164	VUS(4)	LB/VUS	8 b/4 p	0.07	VUS

na—not available; MAF—minor allele frequency; P—pathogenic; LP—likely pathogenic; LB—likely benign; VUS—variant of uncertain significance.

**Table 3 ijms-24-15830-t003:** *MC1R* allelic variants in all probands.

Patient	n° M	Sex	n° Nevi	Group	High-Penetrance Variant	*MC1R* Variants	*MC1R* Allele Type
MH01	19	M	209	FM	None	p.Gln23Ter	u/0
MH02	3	M	4	FM	None	p.Arg163Gln p.Val92Met)	r/r
MH03	1	F	58	FM	None	p.Arg151Cys	R/R *
MH04	3	M	78	MPM	None	p.Arg151Cys p.Val60Leu	R/r
MH05	3	F	23	MPM	None	p.Val60Leu	r/0
MH06	2	F	75	FM	None	p.Arg151Cys	R/0
MH07	4	M	247	MPM	None	wt	0/0
MH08	4	F	97	FM	None	p.Asp294His p.Val60Leu	R/r
MH09	1	M	30	FM	None	p.Asp84Glu	R/0
MH10	1	M	118	FM	None	p.Arg160Trp	R/0
MH11	2	F	65	FM	*CDKN2A* (P) *ACD* (VUS)	p.Val60Leu	r/0
MH12	4	M	203	MPM	None	p.Val60Leu	r/0
MH14	3	M	62	MPM	None	p.Arg160Trp	R/0
MH15	3	M	190	MPM	None	wt	0/0
MH16	6	F	110	FM	*POT1* (VUS)	p.Arg151Cys p.Ser83Leu	R/u
MH17	2	M	167	FM	None	wt	0/0
MH18	3	M	117	FM	None	p.Val60Leu p.Ile264Val	r/u
MH19	2	F	46	FM	None	wt	0/0
MH20	3	F	109	MPM	*BAP1* (P)	p.Arg160Trp	R/0
MH21	4	M	145	MPM	None	p.Arg163Gln	r/0
MH22	2	M	90	FM	None	p.Arg160Trp	R/0
MH23	6	M	59	FM	None	p.Arg151Cys	R/0
MH24	6	F	118	FM	None	p.Arg160Trp p.Ile155Thr	R/r
MH25	2	F	14	FM	None	p.Arg151Cys	R/0
MH26	5	F	178	FM	None	p.Arg151Cys	R/0
MH27	2	F	19	FM	None	p.Arg160Trp	R/0
MH28	2	F	24	FM	*TERT* (VUS)	p.Thr272Met	u/0
MH29	28	F	348	FM	None	p.Arg142His p.Arg151Cys	R/R
MH30	2	F	70	FM	None	p.Val60Leu	r/r *
MH31	1	M	18	FM	None	p.Arg160Trp	R/0
MH32	2	F	7	FM	*ACD* (VUS)	p.Val92Met	r/0
MH33	4	M	316	FM	None	p.Arg151Cys	R/0
MH34	5	F	176	MPM	None	p.Arg142His p.Val60Leu	R/r
MH35	6	M	334	MPM	None	p.Arg151Cys p.Arg163Gln	R/r
MH36	1	F	73	FM	None	p.Arg163Gln	r/0
MH37	2	F	109	FM	None	wt	0/0
MH38	4	M	145	MPM	None	p.Val60Leu p.Gly89Ar	r/u

n° M—number of primary melanomas; M—male; F—female; FM—familial melanoma; MPM—multiple primary melanomas without family history; n° nevi—total nevus count; wt—wild type; * homozygote variant.

**Table 4 ijms-24-15830-t004:** *MC1R* variants frequency—cohort vs. ABraOM control group.

*MC1R* Variant		Cohort (37 Patients)			ABraOM Control Group (609)			Statistics		
		HZ	HT	AF	HZ	HT	AF	OR	95% CI	*p*-Value
**p.D84E**	R	0	1	1.4%	0	2	0.2%	10.93	0.98–122.52	0.052
**p.R142H**	**R**	**0**	**2**	**2.7%**	**0**	**1**	**0.1%**	**43.73**	**3.90–490.07**	**0.002**
**p.R151C**	**R**	**1**	**9**	**14.9%**	**0**	**26**	**2.1%**	**9.25**	**4.34–19.74**	**<0.001**
**p.R160W**	**R**	**0**	**7**	**9.5%**	**1**	**29**	**2.5%**	4.94	**2.08–11.74**	**<0.001**
p.D294H	R	0	1	1.4%	0	21	1.7%	1.04	0.14–7.89	0.969
**Total R**		1	**20**	**29.7%**	**1**	**79**	**6.7%**	**5.94**	**3.44–10.26**	**<0.001**
p.V60L	r	1	8	13.5%	5	107	9.6%	1.41	0.70–2.84	0.333
p.V92M	r	0	2	2.7%	1	46	3.9%	0.69	0.16–2.91	0.611
p.I155T	r	0	1	1.4%	0	7	0.6%	2.36	0.29–19.52	0.425
p.R163Q	r	0	4	5.4%	10	84	8.5%	0.64	0.23–1.79	0.390
Total r		1	15	23.0%	16	244	22.7%	1.02	0.59–1.79	0.924
p.Q23*	u	0	1	1.4%	0	2	0.2%	8.54	0.77–95.33	0.081
**p.G89R**	**u**	**0**	**1**	**1.4%**	**0**	**1**	**0.1%**	**17.08**	**1.06–275.97**	**0.046**
p.I264V	u	0	1	1.4%	0	2	0.2%	8.54	0.77–95.33	0.081
**Total u**		**0**	3	**4.1%**	**0**	**5**	**0.4%**	10.25	**2.40–43.75**	0.002

HZ = homozygotes; HT = heterozygotes; AF = allele frequency; OR = odds ratio; CI = confidence interval.

## Data Availability

Additional data can be available upon reasonable request.

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
