# Peer review of "Characterization of Potential Melanoma Predisposition Genes in High-Risk Brazilian Patients"

_ijms, 2023, doi:10.3390/ijms242115830_

Round 1

Reviewer 1 Report

Comments and Suggestions for Authors

Manuscript could be interesting but the corrections should be done to improve the manuscript.

1.     The justification for the selection of the following genes for analysis: ACD, BAP1, CDKN2A, CDK4, MC1R, MITF, POT1, TERF2IP, and TERT should be provided.

2.     What are the conlusions of the study? Authors missed this part.

3.     What was control group?

4.     What was the concept for the comparison of FM and MPM and skipping the control group?

5.     What was the age range for each group?

6.     The discussion should be improved. Authors should discuss more detailed their results with published data, especially in relation to the main findings (MC1R variants).

7.     Authors should disscuss the basis of the differences between authors' results, "literature and previous Brazilian studies (0-14%), in-220 cluding one from our cancer center (36%, when considering FM + MPM)".

8.     Authors indicated as the limitation the small size of the cohort, but another limitation id lack of control group.Besides the MPM   cohort was very small (only 11 participants).

Author Response

Responses - Reviewer 1:

1.The justification for the selection of the following genes for analysis: ACD, BAP1, CDKN2A, CDK4, MC1R, MITF, POT1, TERF2IP, and TERT should be provided.

We thank reviewer 1 for the careful assessment of our manuscript. Our cohort is composed by melanoma high risk probands, which risk was determined by the number of melanomas cases in their families or number of primary tumors in the individual. Therefore, the genes selected to be studied were the main risk genes for melanoma described to date and within our research funding possibilities.

  1. What are the conclusions of the study? Authors missed this part.

 Our conclusions are described in item 5, right after Materials and Methods (page 18), as determined by the journal's formatting. Please, see below.

 5. Conclusions

A low frequency of high-penetrance genes variants and a high prevalence of MC1R variants found in our cohort emphasize the likely importance of MC1R variants in determining melanoma risk in our population.  The higher frequency of RHC and u MC1R variants in our cohort compared to the control group and also the presence of two MC1R variants, demonstrates that, to a certain extent, the increased melanoma risk of these individuals and families may be related to MC1R genotype combined to pigmentation phenotype, behavior of risk (UVR exposure) and the presence of multiple nevi. Future studies should seek to investigate how MC1R variants interact to other genetic and environmental risk factors in determining melanoma risk among Brazilian melanoma-prone families.”

  What was control group?

 For MC1R analysis, the control group are described in item 4.3 Statistical Analysis (Material and Methods) (page 17). Please, see below. For the other genes, we were not able to perform a case-control analysis considering the small size of patients’ cohort and the rarity of the identified mutations. 

 “A comparison of the MC1R variants frequencies found in our cohort with a control group was performed using logistic regression model to calculate odds ratios (OR) with confidence intervals (CIs) of 95%. The control group used was the database of the Brazilian Online Archive of Variants (ABraOM), which represents a cohort of elderly Brazilians (over 60 years of age) with extensive phenotyping, based on the SABE study census (Health, Wellness and Aging) initiated in 1999, enclosing data from São Paulo city. As approximately 10% of the Brazilian population lives in the metropolitan region of São Paulo, this is considered a representative sample of the national population. It includes 609 individuals, 216 men and 393 women, 517 self-declared white, 156 black, 12 yellow, 2 indigenous, 40 self-declared as “others” and 52 without definition[46] Naslavsky MS, Yamamoto GL, de Almeida TF, et al. Exomic variants of an elderly cohort of Brazilians in the ABraOM database. Hum Mutat. 2017;38(7):751-763. doi:10.1002/humu.23220”

  1. What was the concept for the comparison of FM and MPM and skipping the control group?

 The control group was used only for comparison of the MC1R variants, once our patients showed a high prevalence of MC1R variants and the number of positive patients permitted the statistical analysis.

The comparison between the FM and MPM groups was made in relation to phenotypic characteristics, patients´ demographic data and tumor features, to check for possible differences between high-risk patients for melanoma with and without a family history.

 What was the age range for each group?

 FM – median age at diagnosis = 43.5, range 15-69

MPM – median age at diagnosis = 35.6, range 25-49

These data have been included in Table 1.

  1. The discussion should be improved. Authors should discuss more detailed their results with published data, especially in relation to the main findings (MC1R variants).

We have made substantial changes in the discussion section, removing the more descriptive results, and adding specific paragraphs discussing our findings, including the MC1R results.  

  1. Authors should discuss the basis of the differences between authors' results "literature and previous Brazilian studies (0-14%), including one from our cancer center (36%, when considering FM + MPM)".

This difference is likely to be due to the overall small sizes of ours and previously described cohorts. This information was included in the discussion section.

  1. Authors indicated as the limitation the small size of the cohort, but another limitation is lack of control group. Besides the MPM cohort was very small (only 11 participants).

We appreciate the reviewer's comment, and we acknowledge the limitations of our study. Indeed, the small size of the patients´ cohort and the limited number of participants in the MPM group are important factors that impact the generalizability of our findings and the strength of our conclusions. Regarding the absence of a control group for comparing variants in other genes besides MC1R, the limited number of positive patients impaired any statistical comparisons to ABraOM control population. We understand that the absence of a control group may limit our ability to draw definitive conclusions about the role in melanoma risk of certain genes. We have included these limitations in the discussion section.

Reviewer 2 Report

Comments and Suggestions for Authors

The presented manuscript investigates  germline pathogenic variants in high-penetrance and low- to moderate penetrance genes in Brazilian subjects with possible familiar melanoma (including multiple melanomas).

Overall, the study is well organized and written; nevertheless, there are some issues to correct:

1. Table 1- authors should include gender among variables. Furthermore, age range should be added.

2. Rows 146-148: …the incidence of melanomas located in head/neck and trunk was higher in the MPM group, and the incidence of melanomas in limbs (both upper and lower) was higher in FM patients (p=0.04):"  Is it possible that in FM patients there is a higher frequency of women? Melanomas in the limbs, especially lower limbs, are more characteristic in women. Please clarify.

3. It is unclear how the comparison with AbraOM group was made. Did the authors match data of cases with those from controls, selecting controls with the same ethnicity, gender, and age range of cases? If they did not, it is worth to do so.

4. The discussion seems to be a repetition of the results. Please re-style it.

5. Row 84: UVR concentration. Please change to UVR emission.

Comments on the Quality of English Language

None

Author Response

Responses - Reviewer 2:

  1. Table 1- authors should include gender among variables. Furthermore, age range should be added.

We thank reviewer 2 for the assessment of our manuscript. Table 1 was corrected as requested.

  1. Rows 146-148: …the incidence of melanomas located in head/neck and trunk was higher in the MPM group, and the incidence of melanomas in limbs (both upper and lower) was higher in FM patients (p=0.04):" Is it possible that in FM patients there is a higher frequency of women? Melanomas in the limbs, especially lower limbs, are more characteristic in women. Please clarify.

From 26 FM patients, 10 were male and 16 female. From 11 MPM patients, 8 were male and 3 female. In fact, FM showed a higher frequency of female patients. We have included this observation and the corresponding literature in Discussion. Thank you very much for your contribution.

  1. t is unclear how the comparison with ABraOM group was made. Did the authors match data of cases with those from controls, selecting controls with the same ethnicity, gender, and age range of cases? If they did not, it is worth to do so.

We selected the complete cohort of ABraOM available at the time of the analysis (609 individuals – version SABE609 (hg19)). This cohort represents a cohort of elderly Brazilians (over 60 years of age) from São Paulo city. It includes 609 individuals, 216 men and 393 women, 517 self-declared white, 156 black, 12 yellow, 2 indigenous, 40 self-declared as “others” and 52 without definition. As our patients’ cohort is mostly from the same geographic region (São Paulo) and all ABraOM individuals are over 60 years of age, we considered this database as an appropriate control group representative of the Brazilian population.

  1. The discussion seems to be a repetition of the results. Please re-style it.

We have made substantial changes in the discussion section, removing the more descriptive results, and adding specific paragraphs discussing our findings.

  1. Row 84: UVR concentration. Please change to UVR emission.

Corrected as requested.
